# The impact of changing forest composition in Europe - longest carbon turnover time in unmanaged and broadleaved deciduous forests

Anna Ferretto[1]*, Peter Anthoni[1], Thomas A. M. Pugh[2,3,4], Konstantin Gregor[5], Martin Thurner[1], Carolina Natel[1], David Wårlind[2], Mats Lindeskog[2], Almut Arneth[1,6]

1 Karlsruhe Institute of Technology, Institute of Meteorology and Climate Research, Atmospheric Environmental Research, Garmisch Partenkirchen, Germany, 2 Lund University, Department of Physical Geography and Ecosystem Science, Lund, Sweden, 3 School of Geography, Earth and Environmental Sciences, University of Birmingham, Birmingham, United Kingdom, 4 Birmingham Institute of Forest Research, University of Birmingham, Birmingham, United Kingdom, 5 Technical University of Munich, TUM School of Life Sciences, Freising, Germany, 6 Karlsruhe Institute of Technology, Institute of Geography and Geoecology, Karlsruhe, Germany

* anna.ferretto@kit.edu

**Data availability statement:** The LPJ-GUESS model code, based on svn release 12988 (Pugh et al., 2025) and including the small changes

## Abstract

Forests play a crucial role in Europe's strategy for achieving carbon neutrality. Carbon turnover time - the time that carbon spends in the ecosystem - is a fundamental component in determining forest potential to mitigate climate change. However, there is a significant knowledge gap regarding how current and future forest management practices will affect carbon turnover time. This study aims to compare the effects of various forest management strategies on carbon turnover time in European forests. To achieve this, we used the dynamic global vegetation model LPJ-GUESS to simulate carbon pools and fluxes under stylised forest management scenarios mainly based on changing species composition. We calculated carbon turnover times under two conditions: first, with constant climate and $CO_2$ concentration to assess the isolated impact of forest management; second, under a climate change scenario (SSP3-RCP7.0) to evaluate the combined effects of forest management and climate change. Our results indicate that unmanaged forests and the transition to broadleaved deciduous forests have a similar ecosystem carbon turnover time, which is the longest among all the management options across all the European climatic zones. Climate change decreases ecosystem carbon turnover time in any forest management, in a similar way, especially in cold climates. This study is the first step to include forest management when modelling carbon turnover time and indicates how the shift towards broadleaved forests, which is seen as an important climate-change adaptation strategy in many European regions, can also provide co-benefits for climate-change mitigation.

described in the manuscript, is available under the Creative Commons Attribution 4.0 International through the Zenodo repository at https://doi.org/10.5281/zenodo.17155268 (Anthoni and Wårlind, 2025). The codes used to produce the turnover data and the figures in the manuscript from the raw LPJ-GUESS output are available under the Creative Commons Attribution 4.0 International through the Zenodo repository at https://doi.org/10.5281/zenodo.15269692 (Ferretto, 2025).

**Funding:** This study was made possible thanks to the funds granted to AF by the Alexander von Humboldt-Stiftung. The funders had no role in study design, data collection and analysis, decision to publish, or preparation of the manuscript. TP and ML were funded under the ForestValue programme, the European Commission, VINNOVA, the Swedish Energy Agency and Formas for the project FORECO. This study is a contribution to the Swedish government's strategic research areas BECC and MERGE and the Nature-based Future Solutions profile area at Lund University. KG acknowledges funding from the VELUX Stiftung through the 3FOR-project (project 1897, www.velux-stiftung.ch, www.3for-project.org).

**Competing interests:** The authors have declared that no competing interests exist.

## Introduction

In the second half of the twentieth century, carbon (C) stock in European forests has increased by circa 1.75 times [1] and almost tripled until 2020 compared to 1950 [2]. Expansion of the total forested area explains part of the growth - the dominant driver has been an increase in growing stock volume across existing forests [3]. This increase in stock volume was driven by increasing atmospheric carbon dioxide ($CO_2$) levels (carbon dioxide ($CO_2$) fertilisation) and nitrogen deposition, and also reflects the young age of the forests [4–6]. The age-related growth rate (and biomass carbon sink) will decline in time, but also the impact of carbon dioxide ($CO_2$) fertilisation may diminish. Büntgen et al. (2019) have observed, for example, that faster growth is also associated with earlier tree death [7].

More recently, a decline in C stock has been detected in European temperate forests [8], and at the same time the C sink has been observed to decrease in some parts of Europe [9] and more in general in northern forests [10], also due to drought and its indirect consequences [11]. As stressed by Körner (2017), faster tree growth does not necessarily correspond to greater carbon sequestration: for this to happen, we also need to maintain or increase carbon turnover time in forest ecosystems [12].

Carbon turnover time ($\tau$) refers to the average period in which carbon remains in an ecosystem before being released back into the atmosphere. This metric is hence essential for evaluating the efficiency of carbon storage, its potential contribution to mitigating climate change, and the effects of climate change and/or human (mis)management [13]. Together with the aforementioned carbon dioxide ($CO_2$)-driven change in tree longevity, forest management has a strong influence on $\tau$, through changes in species composition and harvest rates.

Despite its importance, $\tau$ is still an uncertain component of the global forest carbon cycle. Some studies have reported a decrease in turnover time in forests and more generally in terrestrial ecosystems in the recent decades. Yu et al. (2019) found -focusing mainly on mature and unmanaged forests- that across forest climate zones, $\tau$ decreased by more than 2% per year in the period 1955-2018, with small differences between forest types [14]. Similarly, Pugh et al. (2024) reported that increased disturbance due to anthropogenic influence (in the form of land use transition, harvest rate and change in species composition) reduced $\tau$ by 7% and 32% in boreal and temperate forests, respectively [15]. This result resonates with earlier work that highlighted global vegetation models generally diverging substantially in the representation of $\tau$ [16,17], leading to different estimates of projected carbon stock. Carbon residence time in ecosystems has been identified as one of the largest uncertainties in the future estimates of ecosystems' carbon stocks [16,18].

Pugh et al. (2020) compared different Dynamic Global Vegetation Model (DGVM)s to investigate what drives the spatial and temporal differences in vegetation $\tau$ estimates, both currently and in future projections [19]. They depict a very complex picture, where the uncertainty in $\tau$ estimates is not only due to different representations of mortality mechanisms in models, but also caused by different responses in allocation patterns and turnover rates of soft tissues to environmental changes. Furthermore, mechanisms that influence the turnover time act simultaneously at individual, stand, and population levels, and they are complicated to disentangle. Forkel et al. (2019) claim that the uncertainty in Dynamic Global Vegetation Model (DGVM)s' forest carbon estimates could be reduced only with a better representation of the response of $\tau$ to disturbances and extreme events [20]. In addition to disturbances, the consideration of forest management is essential for improving the simulation of forest carbon dynamics [20].

The role of management is increasingly important, especially in European forests, where Senf and Seidl (2021) indicated that disturbance caused by human land use now exceeds that from natural sources [21]. Forest management not only influences mortality, but it also reshapes the structure and the composition of the forests: Noormets et al. (2015) compared two datasets that included managed and unmanaged forest stands and showed that managed forests are almost 50 years younger and contain a significantly higher proportion of coniferous stands (70% vs 53%) [22]. Besides that, they stock about half of the carbon, both in aboveground vegetation and in the soil, compared to unmanaged forests [22]. Including forest management when modelling forest $\tau$ is hence important both for reducing the uncertainties in current model estimates, and also for capturing how both climate change and direct human impacts affect future projections of $\tau$. In Europe, where only 2% of forests are undisturbed by man [23], considering management is of primary importance, especially when exploring future scenarios. This is particularly relevant in view of the EU's goal of climate neutrality by 2050, which relies on forests being a carbon sink [24].

In this study, we used the Dynamic Global Vegetation Model (DGVM) LPJ-GUESS [25–27] to simulate carbon dynamics in current European forests. LPJ-GUESS includes a flexible forest management module, which allowed us to apply stylised management options that we modified from Gregor et al. (2022), focused on changing species composition [28]. One of Pugh et al. (2020)'s hypotheses is that the shifts in forest functional composition drive the response of $\tau$ to environmental changes [19]. In fact, they found that a shift in plant species composition can cause a substantial change in turnover due to population-related mechanisms. To explore this hypothesis further, we compared different forest management routines that shift their current composition and create specific plantations. The general aim is to assess if and how turnover time changes with the different forest management options that we considered. Given the high uncertainties in model estimates of $\tau$, and that the vast majority of European forests are managed, we think this study can contribute to a better understanding of the future of the European carbon sink that is so important for achieving European climate mitigation targets.

## Materials and methods

This study aims to assess the impact of different forest management options on $\tau$ in European forests. We used the Dynamic Global Vegetation Model (DGVM) LPJ-GUESS  [25,26] at a 0.5°x 0.5°  spatial resolution to simulate ecosystem carbon pools and fluxes. We conducted two sets of simulations: (a) management-only simulations, with constant atmospheric carbon dioxide ($CO_2$) and nitrogen deposition and fixed climate; and (b) management and climate change simulations, applying management scenarios combined with climate change, atmospheric carbon dioxide ($CO_2$) and nitrogen deposition projections (see Input data). In both sets of simulations, from 2010 we applied our range of stylised forest management options. We will refer to these two different sets of runs as "management-only simulations" and "management and climate change simulations", respectively. We calculated $\tau$ in the ecosystem, stem and soil, with a closer look at some relatively long-term soil pools.

### Model description

LPJ-GUESS has previously been applied at both regional and global scales, and uses input data of climate, soil, nitrogen deposition, and atmospheric carbon dioxide ($CO_2$) concentrations to simulate plant growth and competition among different PFTs [26]. The model returns information about the carbon and nitrogen cycles, water fluxes, and forest dynamics. In this study, we used LPJ-GUESS svn release 12988, based on Pugh et al. (2025) [29].

We have added the detailed reporting of carbon fluxes between all soil pools as an additional output and updated the mechanism by which nitrogen limitation affects Soil Organic Matter (SOM) decomposition (S1 Appendix). This version also includes a forest management module [27], which allows the simulation of a range of management options, including clear-cuts, thinning events, and the selection of species to plant. We also included a parametrisation for European species [27,30] to model the main European plant species instead of more generic PFTs. Some parameters were also changed according to Pugh et al. (2025) [29].

## Tree mortality

In LPJ-GUESS, tree mortality is modelled as a stochastic process, where each tree has an annual probability of dying due to factors such as aging, stress (which reduces the growth efficiency) or self-thinning. Mortality can also occur if bioclimatic limits for a specific PFT fall outside their range [19]. Besides these mortality mechanisms, trees can also die because of a recurring disturbance, fire, or because they are harvested. The disturbance is implemented in LPJ-GUESS as a stochastic patch-destroying process with a specified average return interval (400 years in this study, following Lindeskog et al. (2021) [27]). Each forest stand is modelled with a defined number of patches or samples (25 in this study), which are randomly destroyed by disturbance based on the disturbance return interval. Fire was modelled using the SIMFIRE-BLAZE model [31,32], but was turned off for managed forests (with the assumption that in managed forests the risk of fires can be greatly reduced if proper fire mitigation strategies are taken [33]). In LPJ-GUESS, when trees are harvested, 66% of wood biomass (stem, twigs and coarse roots) and 30% of leaf biomass are removed from the stand, while the rest remains in the ecosystem as litter. The whole removed leaf biomass and 67% of the removed wood biomass are oxidised and lost to the atmosphere in the same year, while the rest goes into a product pool with an oxidation rate of 4% per year [27]. Harvest occurs through clear-cuts or thinning events, described further in the following section.

## Forest management and settings

In this study, management options refer to changing species composition to create specific plantations and are based on Gregor et al. (2022) [28]. They are the following:

- baseline (base): every time that trees die (for natural causes or because they are harvested), the same species as before are replanted;
- transition to needle-leaved evergreen (toNE): every time that trees die, only needle-leaved evergreen species are replanted;
- transition to broadleaved deciduous (toBD): every time that trees die, only broadleaved deciduous species are replanted;
- transition to broadleaved evergreen (toBE): every time that trees die, only broadleaved evergreen species are replanted;
- transition to unmanaged (unmanaged): from 2010 onwards, no management is applied anymore. Before 2010, the forest undergoes the same initialisation process as the other options (see details below).

Trees are replanted with a lag of one year, unless the reason for the death is a clear-cut, in which case they are replanted the same year.

Although defining a single forest management scenario that applies uniformly across all of Europe is not realistic *per se*, it allows us to investigate our primary research question (*i.e.*

how changing species composition affects $\tau$) in different climatic zones. The stylised replanting scenarios could nevertheless reflect strategies relevant to different regions. Replanting only needle-leaved species simulates widespread practices in boreal and temperate Europe, where large areas have historically been converted to conifer-dominated (often monoculture) stands for timber production [34]. Replanting broadleaved deciduous species represents a shift toward more climate-resilient, structurally diverse forests, promoted in the EU Forest Strategy for 2030[35]. Meanwhile, the use of drought-tolerant, native broadleaved evergreen species is suggested in Mediterranean regions as an adaptation strategy to climate change [36].

We applied the management settings on top of an 80-year clear-cut rotation cycle, with the first clear-cut occurring in 2010 (except in the unmanaged forests). To facilitate comparison across different management routines, clear-cut events were synchronised to occur in the same year for all the managed options. Forest rotation length depends on many factors, such as national legislation, forest type, and management goals. Our choice of 80 years was hence arbitrary, but it falls within the range commonly applied in Europe and is reasonable for both needle-leaved and broadleaved species [37,38]. In between clear-cuts, we also introduced thinning events, regulated by the self-thinning Reineke's rule [39], described in Lindeskog et al. (2021) [27]. This automated thinning procedure aims to avoid the self-thinning, and is parameterised differently between needleleaved and broadleaved species [27], to reflect the fact that, compared to needleleaved species, broadleaved species, tend to have lower maximum stand density at which self-thinning is triggered [40]. Further simulations with a limited amount of random grid-cells (10 for each climatic zone) were run with a rotation period of 60 years and 100 years, to check whether and in which terms, the clearcut interval influences the results ( S4 Appendix).

To initialise the model with a realistic representation of current European forests, we systematically clear-cut and replanted some stands every 10 years between 1871 and 2010, following Lindeskog et al. (2021) [27]. In 1870, the entire forest area was considered "pristine", then every 10, in each grid cell some stands were clear-cut and replanted and became "managed" forest. The timing and extent of these clear-cuts were set so as to reproduce the observed 2010 age distribution from the Global Forest Age Dataset (GFADv1.0, [41]). This process continued until 2010, when the majority of the European forests have been converted to managed, and only the forest fraction corresponding to "forests older than 140 years" in the GFAD, has been left untouched. The forest management options and the forest initialisations were the same for both the management-only and the management and climate change simulations.

Since the different types of forests do not necessarily grow everywhere in Europe, we restricted our analysis to the grid cells with an average tree height of 5 meters, which is generally considered one of the thresholds to define a forest [42,43], and allows us to check if a certain PFT can grow in a determined climate.

## Input data

Daily temperature, radiation, precipitation, wind and relative humidity data at 0.5°× 0.5° resolution were obtained from the ISIMIP3b (Inter-Sectoral Impact Model Intercomparison Project) bias-adjusted data of the MPI-ESM1-2-HR Global Climate Model (GCM) [44], and atmospheric carbon dioxide ($CO_2$) concentration data from the global carbon project [45]. Monthly nitrogen deposition data were taken from the Coupled Model Intercomparison Project 6 (CMIP6) ensemble [46]. The forest cover and structure were obtained from the GFAD [41]. From 2010 on, the pristine and managed forest areas were held constant, but the beginning of forest management starts modifying forest structure (through clear-cuts and

thinning) and species distribution (applying different planting systems). Our analysis was applied to the managed forest fraction only. We performed 10 model runs (5 management options for the management-only simulations, and 5 management options for the management and climate change simulations). A brief description of the runs and the data used in each of them are given in Table 1 and the text below.

The management-only simulations began with a 1200-year spin-up to initialise species composition and soil and plant carbon pools. For this spin-up, the detrended 1995–2014 climate was recycled. carbon dioxide ($CO_2$) concentration and nitrogen deposition of the year 2014 were also prescribed [45]. The spin-up period ends in 2010, after which the management starts, but the same settings for climate, nitrogen deposition and carbon dioxide ($CO_2$) concentration are maintained. In the management and climate change simulations, we used the climate and nitrogen projections from the CMIP6 ensemble for 1850-2100, forced by the SSP3-RCP7.0 (Shared Socioeconomic Pathway 3 - Representative Concentration Pathway 7.0) scenario [46]. Atmospheric carbon dioxide ($CO_2$) concentration data for 1850-2100 consistent with the CMIP6 GCM forcing were used. During a 1200-year spin-up, the detrended 1850-1879 climate was recycled, and the 1850 carbon dioxide ($CO_2$) concentration and nitrogen deposition data were used. We chose SSP3–RCP7.0 to represent a high but plausible emissions pathway. Although European climate policies aim to keep warming below 2°C, Europe contributes only a small share of global emissions, so insufficient mitigation elsewhere could still push global temperatures well beyond this target. Following Huard et al.(2022) [47], who found SSP5–RCP8.5 unlikely in the second half of the 21$^{st}$ century due to fossil fuel constraints, SSP3–RCP7.0 offers a severe yet realistic trajectory, making it well suited for assessing upper-bound climate impacts on European forests. A smaller number of grid cells (10 in each climatic zone) was also tested with a lower-forcing scenario (SSP1-RCP2.6), to check if different levels of climate warming could lead to differences among the management options included (S4 Appendix).

## Calculation of $\tau$

We calculated $\tau$ as the ratio between the carbon stock of each pool and the carbon flux that leaves that pool, following Pugh et al. (2020)[19]. This approach assumes that the pool is in equilibrium or in a so-called "steady state" (*i.e.* over a defined time, carbon input equals carbon output). This is often not the case, because forests are very dynamic, and the processes that control their functioning are influenced by many factors (e.g. climate, disturbance, management), which constantly modify the size of the forest carbon pool. Nevertheless, many

**Table 1**. Data used in the simulations.

| Scenario | Climate | carbon dioxide ($CO_2$) and nitrogen deposition | Forest structure |
|---|---|---|---|
| Management only | detrended and recycled 1995-2014 data | 2014 | Spin-up and 1850-1870: potential natural vegetation; 1870-2010: land use and stand type files that reproduce the forest structure of 2010 from [41]; After 2010: constant value of 2010 |
| Management and climate | Spin-up: detrended 1850-1879 data; 1850-2100: CMIP6 data and projections | Spin-up: value of 1850; 1850-2100: CMIP6 data and projections | Spin-up and 1850-1870: potential natural vegetation; 1870-2010: land use and stand type files that reproduce the forest structure of 2010 from [41]; After 2010: constant value of 2010 |

studies have relaxed this assumption and used 'steady state' equations also for transient trajectories, arguing that in a long-term perspective, the variability in the fluxes is much smaller than the size of the carbon stocks [48]. We also tested the steady-state assumption by comparing our results with $\tau$ calculated as carbon pool divided by the flux that enters the ecosystem (see S2 Appendix).

In the management-only simulations, we kept climate, carbon dioxide ($CO_2$) and nitrogen deposition constant, thereby isolating the effect of forest management on $\tau$. We ran the simulations for 240 years after spinup, to have at least three vegetation cycles (80 years for each cycle) and to leave time to mitigate the effect of the forest initialisation. We calculated the mean carbon (C) pool and the mean of the sum of the fluxes that left the corresponding C pool for the last 30 years of the forest cycle, to get the mean $\tau$ for that period. We chose the last 30 years of the cycle to let the forest grow after the previous clearcut but to have a long enough period to avoid possible fluxes anomalies in single years. A comparison between *base* and *unmanaged* gives information about the effects of the clear-cut and the thinning events on $\tau$. On the other hand, comparing the baseline to the other management options helps us understand how species composition affects $\tau$.

In the management and climate change simulations instead, we calculated $\tau$ over the period 2060-2089, which corresponds to the last 30 years of the first rotation cycle (right before the clear cut).

The ecosystem C turnover time ($\tau_{\text{eco}}$) was obtained by dividing the ecosystem C pool ($C_{\text{eco}}$) by the sum of the fluxes that leave the ecosystem each year ($F_{\text{turn eco}}$).

$$\tau_{\text{eco}} = C_{\text{eco}}/F_{\text{turn eco}}$$

(adapted from [49])

$C_{\text{eco}}$ represents the total C pool, which is made of the above-ground C (stem and litter) and the soil C.

The C fluxes that leave the ecosystem are:

$$F_{\text{turn eco}} = F_{R_a} + F_{R_h} + F_{\text{DOC}} + F_{\text{fire}} + F_{\text{harv to products}} + F_{\text{harv to atm}}$$

where $F_{R_a}$ is the autotrophic respiration, $F_{R_h}$ is the heterotrophic respiration, $F_{\text{DOC}}$ is the C flux lost as Dissolved Organic Carbon (DOC), $F_{\text{fire}}$ is the C lost because of fires, and $F_{\text{harv to products}}$ and $F_{\text{harv to atm}}$ are the C fluxes that leave the ecosystem as a consequence of the harvest, and end up in the product pool and in the atmosphere, respectively.

In the above-ground vegetation, there are pools with very different $\tau$ (stem *vs* leaves and roots). We focused on stem, branches and coarse roots (hereafter called "stem"), which have the longest $\tau$, and we ignored the leaf and fine-root pools, which in comparison have an ephemeral $\tau$. The stem carbon turnover time ($\tau_{\text{stem}}$) was calculated as the ratio between the C stored in the stem ($C_{\text{stem}}$) each year and the sum of C fluxes ($F_{\text{turn stem}}$) that are lost by the stem C pool at the end of the year due to mortality, harvest, fire or disturbance.

$$\tau_{\text{stem}} = C_{\text{stem}}/F_{\text{turn stem}}$$

(adapted from [49])

The flux that leaves the stem C pool, can be decomposed in:

$$F_{\text{turn stem}} = F_{\text{mort}} + F_{\text{dist}} + F_{\text{fire}} + F_{\text{harv}}$$

(adapted from [19])

where $F_{\text{turn mort}}$ is the outer flux of C lost through physiological mortality processes, $F_{\text{dist}}$ and $F_{\text{fire}}$ are the fluxes of C lost due to mortality caused by disturbance and fires, and $F_{\text{harv}}$ is the flux that leaves the stem pool as a consequence of the harvest (and goes into harvest products, into the litter, or is lost to the atmosphere).

The turnover time in the soil ($\tau_{\text{soil}}$) was calculated following the same approach, dividing the soil carbon pools ($C_{\text{soil}}$) by the fluxes that leave that pool every year ($F_{\text{soil}}$):

$$\tau_{\text{soil}} = C_{\text{soil}}/F_{\text{soil}}$$

We first considered the soil as a single pool. In this case, $F_{\text{soil}}$ corresponds to the total heterotrophic respiration ($F_{\text{R}_\text{h}}$). In LPJ-GUESS, soil C pools and fluxes are modelled using the Century model [50], summarised in Fig. 1. The soil pools represented in the model have an enormously different $\tau$, which can span from less than a year (in the metabolic pools) to thousands of years (in the passive SOM pool). To achieve a more detailed analysis of soil C dynamics, we analysed some pools separately: the surface coarse woody debris pool ($\tau_{\text{surfcwd}}$), the surface fine woody debris pool ($\tau_{\text{surffwd}}$), the surface humus pool ($\tau_{\text{surfhum}}$), and the long SOMpool ($\tau_{\text{slow}}$), which have a $\tau$ in the order of decades to around a century (equations in S3 Appendix).

In a very few grid cells, the fluxes that left the pool (ecosystem, stem or soil), were very low, resulting in high $\tau$ values compared to the rest of Europe. To avoid distortions, we used the 99% interquartile range in all the calculations, excluding these extreme values.

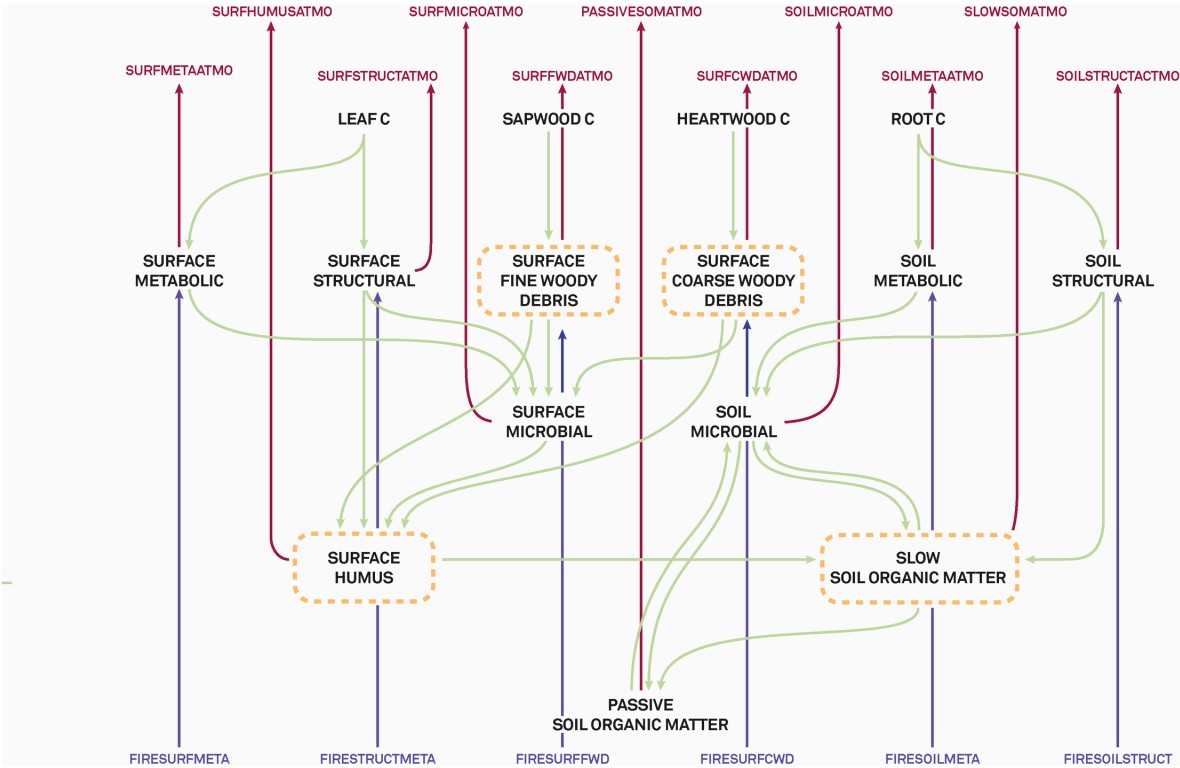

**Fig 1. Soil carbon pools and fluxes modelled in LPJ-GUESS.** Carbon pools are shown in black. Carbon fluxes are represented by the arrows (green: between soil pools, red: from the soil pools to the atmosphere, blue: from burnt vegetation to the soil)

It is also important to note, that in a harvest setting, which potentially kills the trees before their natural mortality, harvest is a large component of the outflows and influences the absolute values of $\tau$ (in particular $\tau_{stem}$, and to a lesser extent $\tau_{eco}$). Here, we calculated $\tau$ as the average of the years 50-79 after a clearcut, in which the harvest flux is only in the form of thinning. Including a clearcut in our calculation would have made $\tau$ shorter. Given the arbitrariness in this choice (we chose not to include a clearcut flux to focus solely on the vegetation shift), we do not focus on the absolute values of $\tau$, rather we will discuss the differences between the management options.

## Climatic zones

To compare mean $\tau$ values between forest management options, we divided Europe into different climatic zones, according to the Köppen-Geiger classification. Data at 0.5°resolution were obtained from Beck et al. (2023) [51]. We grouped some classes since some climatic zones were present in Europe in very few grid cells. Our final classification includes the following classes: "arid", "cold with cold summer", "cold with warm summer", "temperate without dry season", "temperate with dry summer" and "tundra". A map of the climatic zones can be consulted in S11 Fig. Since the "tundra" region encompasses a very small number of grid cells (and in most of them forests do not reach a 5-meter threshold), we excluded it from our analyses.

## PFT transition

We also investigated the impact of the actual change in PFTs on C $\tau$. We calculated the average $\tau$ in the grid cells with a previous dominance of broadleaved and of needle-leaved PFTs, for each management option. Since needle-leaved deciduous and broadleaved evergreen forests are dominant only in a very limited number of grid cells (S12 Fig), we only distinguished between the broader groups of "needle-leaved" and "broadleaved" forests.

## Results

### Management-only simulations

The simulation of different management options across Europe revealed diverse impacts on $\tau_{eco}$ compared to the baseline, but within the management options, the patterns were spatially quite uniform. Compared to the baseline, $\tau_{eco}$ in the unmanaged forests is longer across Europe, except in the northernmost parts of Sweden and Finland (Fig 2a). There is very small difference between the baseline and the toNE forest in most of Scandinavia, Ireland, the Czech Republic and the Iberian peninsula (Fig 2b) (where also before the beginning of the management, needle-leaved species are dominant (see S12 Fig)), while everywhere else, the baseline has a slightly longer $\tau$. As for the unmanaged forest, the toBD transition also shows a longer $\tau_{eco}$ all around Europe compared to the baseline (Fig 2c), with a south-north gradient. Finally, $\tau$ in the case of a transition to BE forests is slightly shorter than in the baseline across Europe (Fig 2d).

Plots of relative changes in $\tau$, between the management options are shown in S13 Fig.

With the exception of Norway, Sweden, Finland and Ireland, $\tau_{stem}$ in the unmanaged forest exceeds the baseline values across Europe (Fig 2e). The transition to NE does not greatly differ from the baseline in the northern countries, Ireland and most of the Iberian peninsula, while it shows a longer $\tau$ in the United Kingdom and central and eastern Europe (Fig 2f). $\tau_{stem}$ in the toBD management option are substantially shorter compared to the baseline case in the northern countries and Ireland (Fig 2g) but do not greatly differ from the baseline in the

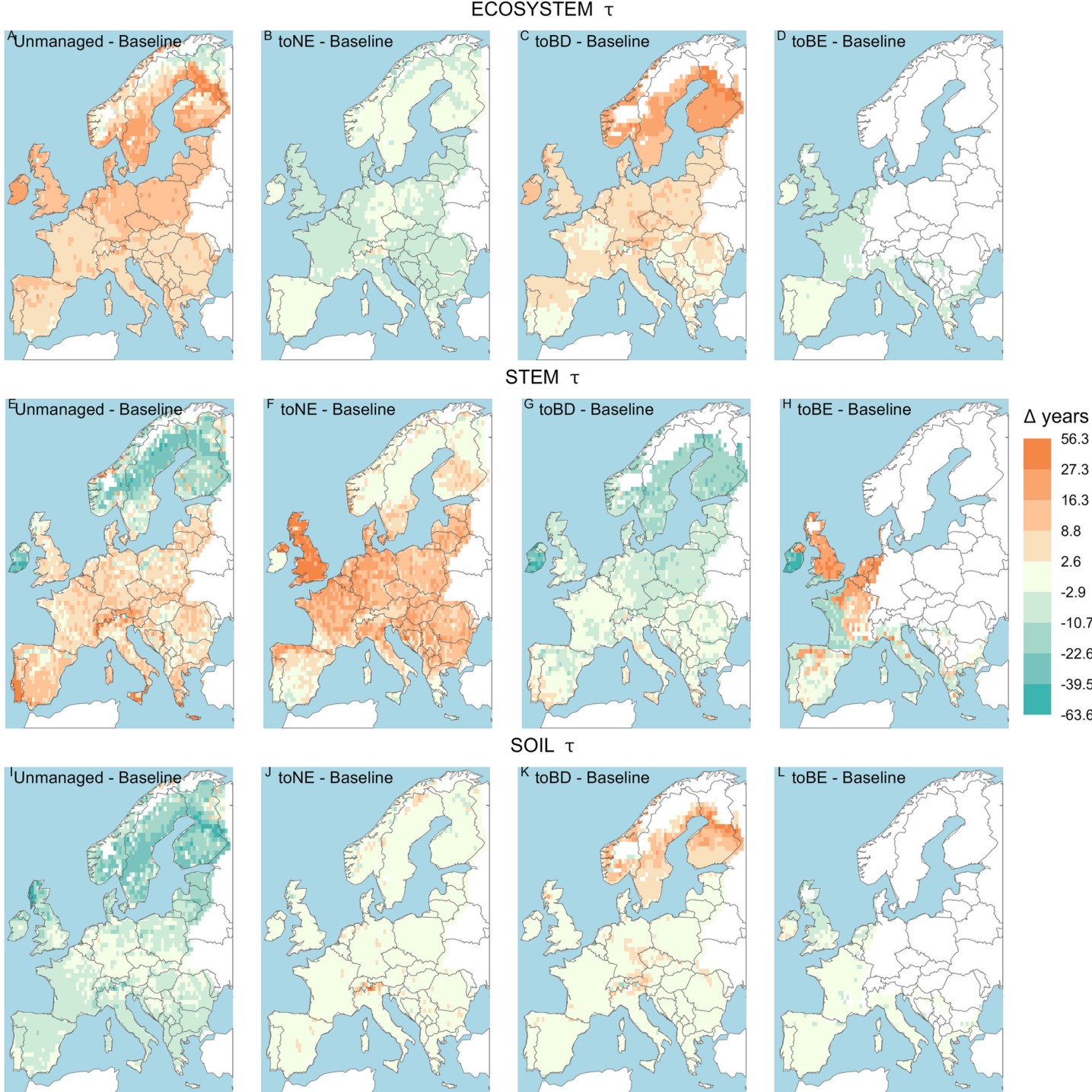

**Fig 2. Differences in $\tau$ between each management option and the baseline scenario, in the management-only simulations - 30-year average before the last clear-cut.** Different shades of green indicate that $\tau$ is longer in the baseline than in the other management options, while different shades of orange indicate that it is shorter. Areas where the forest does not reach an average tree height of 5 meters are excluded

rest of Europe. We also found a contrasting impact when transitioning to BE forests, in that $\tau_{stem}$ becomes shorter in Ireland and eastern France, while it is longer in the Netherlands, Belgium, the United Kingdom, northern Spain and northern and central France (Fig 2h), all areas dominated by broadleaved forests (S12 Fig).

While $\tau_{eco}$ and $\tau_{stem}$ are overall longer in the unmanaged forest compared to the baseline, for soil turnover times we find the opposite: $\tau_{soil}$ is shorter almost everywhere when the forest is changed to unmanaged, especially so in northern Europe and the Alps (Fig 2i). Differences between the baseline and the different plantation systems were much smaller in general but still substantial in some areas (e.g. longer $\tau_{soil}$ in the Alps in the transition to NE forests (Fig 2j) and in the northern countries in the transition to BD forests (Fig 2k)).

Fig 3 summarises across Europe which management option results in the longest $\tau$.

$\tau_{eco}$ of the unmanaged forest was longest almost everywhere (Fig 3a). Exceptions are the northermost part of Finland and Sweden, where the baseline option returned the longest $\tau_{eco}$, and the central area of Sweden, southern Norway and Finland, and the Alps, where broadleaved deciduous forests showed the longest $\tau_{eco}$. Excluding the unmanaged forest, *i.e.* looking only at the baseline and vegetation transition to monocultures, suggests that broadleaved deciduous forests have the longest $\tau_{eco}$ across Europe (Fig 3b). Special cases are northern Finland and the mountain region between Norway and Sweden (where the baseline option shows the longest $\tau_{eco}$), and some areas of the Alps (where the longest $\tau_{eco}$ is observed in the needle-leaved evergreen forests). The longest $\tau_{stem}$ was simulated in most of Europe in the transition to the needle-leaved evergreen forest (Fig 3c). The toBE management option prevails in the Netherlands, Belgium, northern France and Spain, and the baseline in Ireland and scattered grid cells in Norway, Sweden and Finland. In the case of the stem, the only areas where $\tau$ is the longest in the unmanaged are southern Spain, Portugal, Italy and Greece, and some mountain areas (Alps, Pyrenees, Scandinavian mountains). Here, when the unmanaged forest is not considered (Fig 3d), the longest $\tau_{stem}$ is observed in the transition to NE (on the mountains) and to BD (in the remaining areas). In the soil, $\tau$ in managed forests always exceeds the values of the unmanaged case (Fig 3e). In cold climates with both cold or warm summers, the transition to BD forests shows generally the longest $\tau_{soil}$ (Scandinavia and Eastern Europe). The toNE option has the longest $\tau_{soil}$ in areas with temperate climates without dry season (except in Ireland, where toBE prevails) and in the arid climate. In temperate climates with dry summer, the toBE forest returns the longest $\tau_{soil}$.

## Management and climate change simulations

The patterns of $\tau_{eco}$ in the management and climate change simulations follow those of the management-only simulations: transition to unmanaged forests and broadleaved deciduous forests show a longer $\tau_{eco}$ across Europe (Fig 4a, 4c). The transition to NE forests and BE forests leads to a slightly shorter $\tau_{eco}$ across Europe (Fig 4b, 4d).

As for the management-only simulations, also in the climate and managed simulations $\tau_{stem}$ is much more varied than $\tau_{eco}$. The unmanaged forest is projected to have a much shorter $\tau_{stem}$ in northern and central Europe (especially in the northern countries and in Ireland), but a longer $\tau_{stem}$ in the south (Iberian peninsula, Italy and Greece) (Fig 4e). The same pattern is observed in the transition to BD forests (Fig 4g), although with much smaller differences in southern and central Europe. In terms of patterns, transitioning to NE forests does not greatly differ from the climate-only simulations (Fig 4f), but the intensity of the difference is much higher: $\tau_{stem}$ is simulated to be much longer across Europe, especially in Great Britain and in eastern Europe. Finally, a transition to BE forests produces scattered results, with longer $\tau_{stem}$ in Scotland, the south of the Iberian peninsula, parts of Denmark and eastern

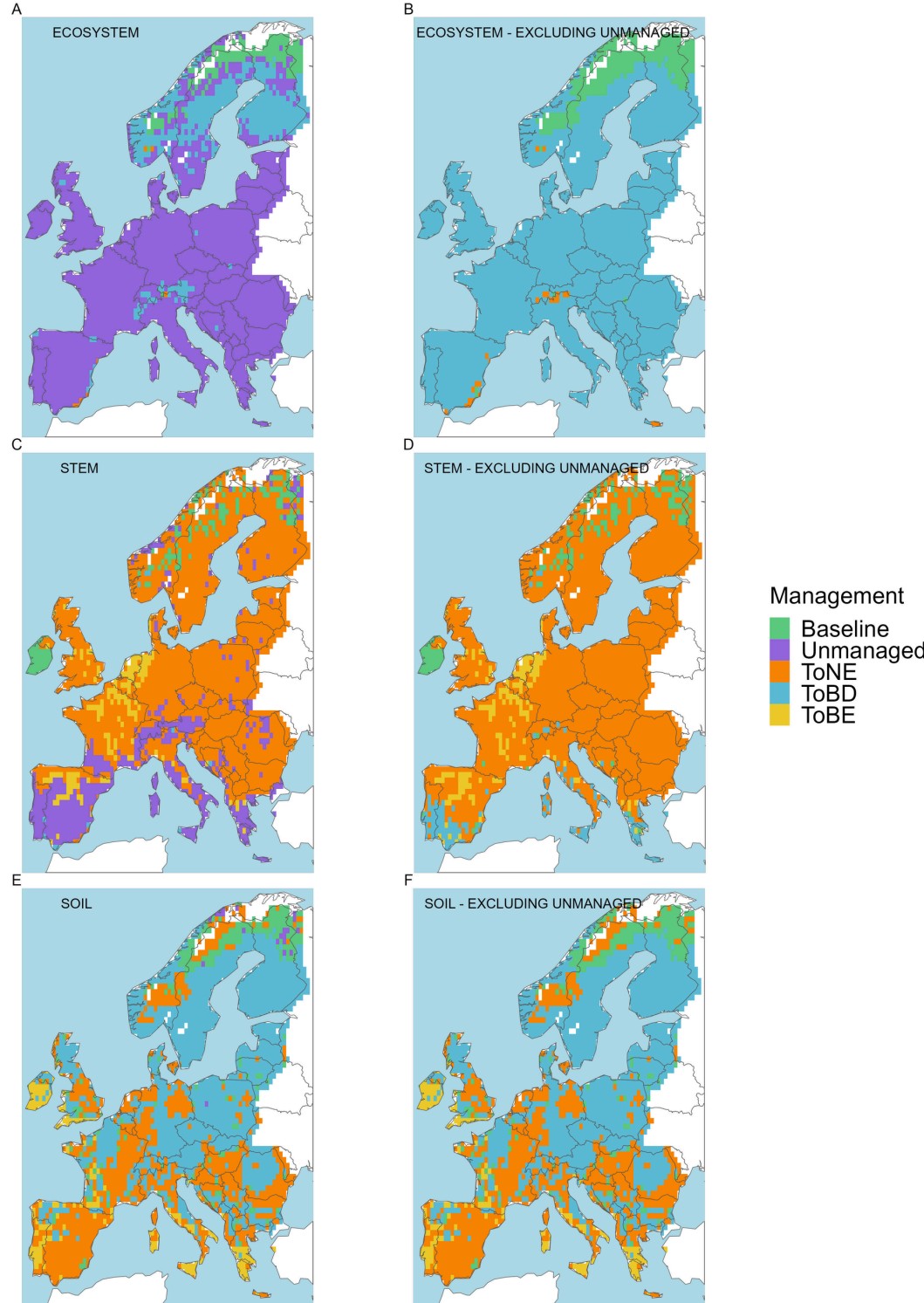

**Fig 3. Management option with the longest mean $\tau$.** A: in the ecosystem; B: in the ecosystem, excluding the unmanaged option; C: in the stem; D: in the stem, excluding the unmanaged option; E: in the soil; F: in the soil, excluding the unmanaged option. The mean refers to the last 30 years of the 3$^{rd}$ management cycle, excluding the final clear-cut

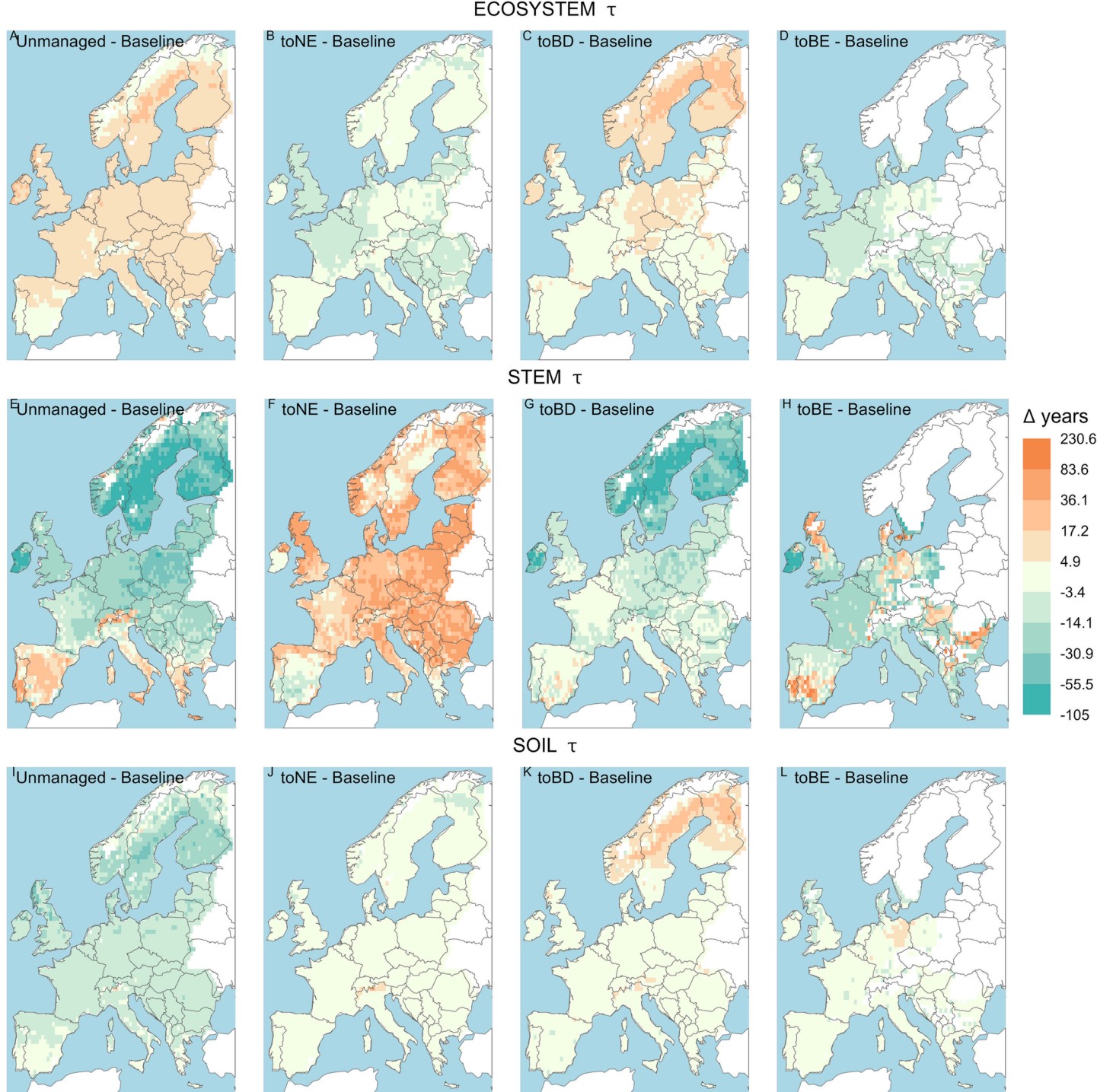

**Fig 4. Differences in $\tau$ between each management option and the baseline scenario, in the management and climate change simulations - 30-year average before the last clear-cut.** Colour coding as in Fig 2. Areas where the forest does not reach an average tree height of 5 meters are excluded.

Germany, Hungary and an area between Romania and Bulgaria, and shorter $\tau_{stem}$ everywhere else (Fig 4h).

$\tau_{soil}$ follows again the same pattern observed in the management-only simulations. In the transition to unmanaged forests, $\tau_{soil}$ is shorter across Europe, especially in the northern countries and Scotland (Fig 4i). No great differences are observed in the transition to NE forests, with only a slight decrease in northern Finland, and an increase in the Alps, compared to the baseline option (Fig 4j). In the transition to BD forests, a great increase is observed in the northern countries, especially in northern Finland and Sweden, while everywhere else there are no great differences with the baseline (Fig 4k). In the transition to BE forests, $\tau_{soil}$ is generally predicted to get shorter, with only some areas in Germany and Hungary showing a slightly longer $\tau_{soil}$ (Fig 4l). Plots of relative changes in $\tau$ for the management and climate change simulations between the management options are shown in S14 Fig.

## $\tau$ variations across climatic zones

The mean $\tau$ was calculated for each climatic zone and each management option (Fig 5), to test whether broad climate patterns affect how $\tau$ responds to the stylised management cases (for the values refer to S1 Table). The climatic zones the analysis is based on, refer to the period 1991 - 2020 in [51] (S11 Fig, panel A).

In the management-only simulations (Fig 5a), unmanaged forests and BD forests always show the longest $\tau_{eco}$ irrespective of the climatic zone and with small differences between these two options. The shortest $\tau_{eco}$ is, instead, always observed in evergreen forests (broadleaved evergreen in the climates where they grow, and needle-leaved evergreen in the cold climates). The baseline option and the NE forest generally have similar $\tau_{eco}$, with the greatest differences (4 years longer in the baseline) in the cold climate with warm summer and the temperate climate without dry seasons. This clear pattern in $\tau_{eco}$ is explained both by higher total C pools and lower outflows in unmanaged and BD forests, while evergreen forests, especially BE, show the lowest pools and at the same time, higher outflows, particularly due to high autotrophic respiration ( S15 Fig and S16 Fig). The only exception is in the cold climate with cold summers, where unmanaged forests store the lowest total carbon. When climate change is included (Fig 5b), $\tau_{eco}$ is always reduced, with the highest decrease in cold climates with cold summers (between 20 and 30 years shorter in all the management options considered). Despite the visible decrease in $\tau_{eco}$, climate change does not substantially modify the order between the forest management options, with unmanaged and BD forests being still those with the longest $\tau_{eco}$ and the BE forests with the shortest, together with NE forests. In general, the transition to BE forest, despite remaining the management option with the shortest $\tau_{eco}$, has the smallest impact from climate change (Fig 5b, 5c). In S17 Fig and S18 Fig, we observe that the decrease in $\tau_{eco}$ is caused by an increase in the outflows. In fact, while the total C pool tend to increase or stay stable in most of the management-climate combinations (with a few exceptions with a slight decrease), we notice a strong increase in autotrophic respiration (particularly evident in the cold climates, especially in the baseline and NE options). Heterotrophic respiration is generally stable across management options, with a few exceptions. In cold climates, it increases under all management options, though most strongly in unmanaged forests. Beyond this, unmanaged stands also show consistently increasing heterotrophic respiration across all climates, with the strongest intensification in cold regions. In these same regions, unmanaged forests are also characterized by higher fire-related fluxes.

In contrast to $\tau_{eco}$, for what concerns $\tau_{stem}$ there is not a clear "management ranking" because here the influence of the type of climate on the different management options is greater, leading to a much more diverse outcome. In the management-only simulations

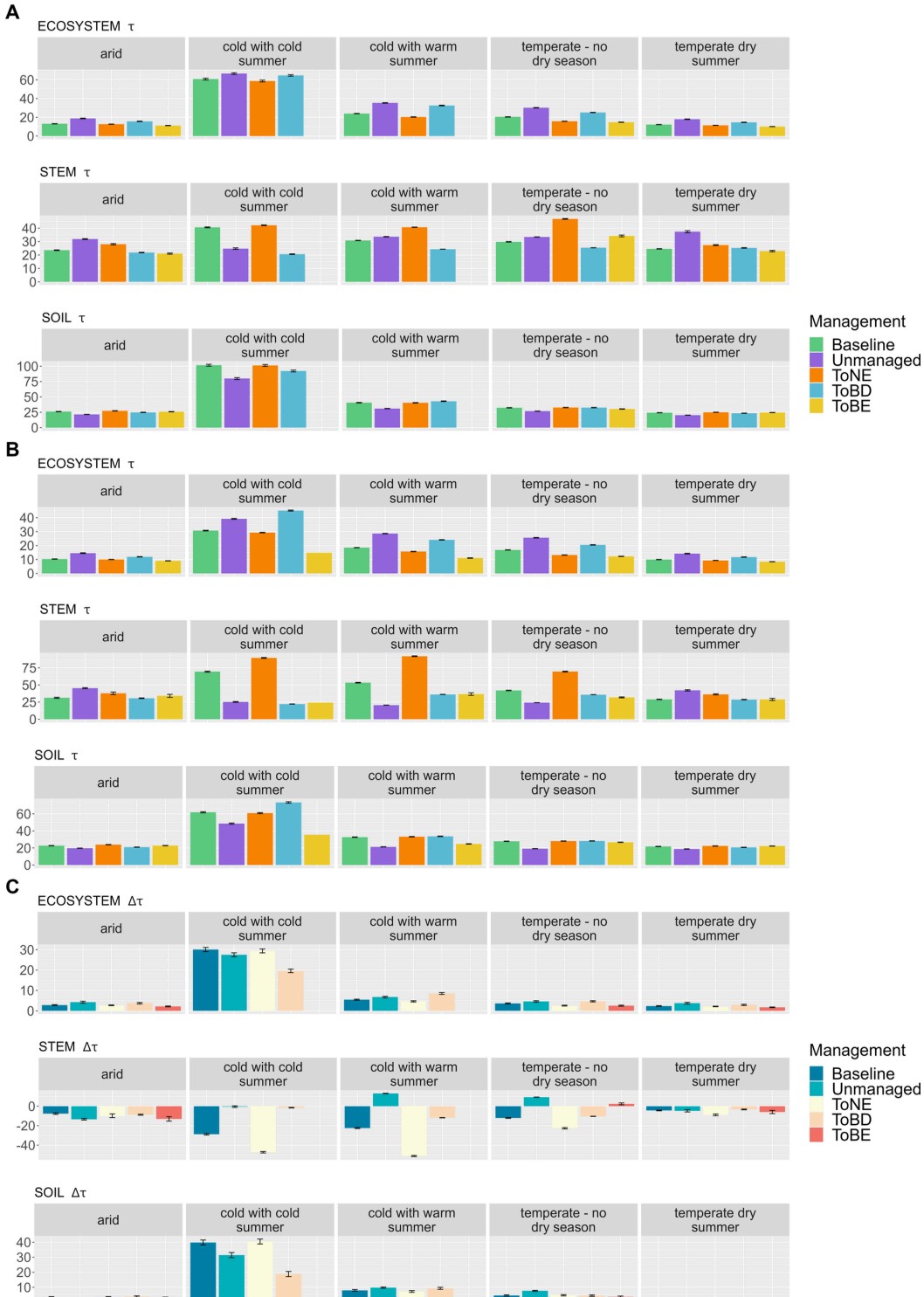

**Fig 5. Mean $\tau$ (with standard error bars) for each management type, in different climatic zones, for the ecosystem, the stem pool and the soil pool.** A: in the management-only simulations; B: in the management and climate simulations; C: difference between A and B. The mean refers to the last 30 years of the $3^{rd}$ management cycle, excluding the final clear-cut

(Fig 5a), transitioning to NE forests produces the longest $\tau_{stem}$ in cold climates and temperate climates without a dry season, while unmanaged forests have the longest $\tau_{stem}$ in arid and temperate climate with dry summers. The baseline has a similar $\tau_{stem}$ to NE forests, except in cold climates with warm summers and temperate climates without dry seasons, where $\tau_{stem}$ is 10 and 17 years shorter. A closer look at the stem C pools and fluxes (S15 Fig and S19 Fig) reveals that unmanaged forests tend to have higher C pools compared to the other management options, in all the climates except in the cold climate with cold summers, where they are equivalent. At the same time, unmanaged forests show a higher outflow due to a higher mortality and, in cold climates, to fire contribution. On the contrary, the NE forests have in general a lower stem C pool compared to the unmanaged forests, but their outflows are among the lowest in all the climate. In particular, mortality losses are so much lower compared to the unmanaged forests, that they are not compensated by the harvest losses. Broadleaved forests always show the shortest $\tau_{stem}$, with small differences between deciduous and evergreen, except in the temperate climate without dry season, where the BD forests have a $\tau_{stem}$ around 10 years shorter than the BE forests. In this case, compared to NE forests, the main outflow is due to harvest. This reflects the different thresholds that trigger the thinning in our simulations (see Forest management and settings): because broadleaved species reach self-thinning thresholds at lower stand densities than conifers, thinning is triggered more frequently, resulting in higher harvest losses.

The climatic zones with the longest $\tau_{stem}$ are primarily found in temperate climates: temperate without dry seasons for the NE, BE and BD forests, and temperate with dry summers for the unmanaged forests. The baseline, instead, has the longest $\tau_{stem}$ in cold climates with cold summers. In the management and climate change simulations (Fig 5b), we observe the opposite trend compared to what happened to $\tau_{eco}$, with longer values of $\tau_{stem}$ in all the climates and for all the management options, except for the unmanaged forests in the cold climate with warm summers and the temperate climate without dry season, where $\tau_{stem}$ decreases of 13 and 9 years, respectively (Fig 5c). The management option with the greatest increase in $\tau_{stem}$ is mostly the transition to NE forests, where in cold climates it is predicted to be around 50 years longer than in the management-only simulations, followed by the baseline (over 20 years) in the same climatic zones. In temperate climates without dry season, the pattern of $\tau_{stem}$ is the same, just with a smaller difference (12 years longer for the baseline and 23 for the transition to NE forests). The arid climate is the only area where the greatest increase in $\tau_{stem}$ is instead observed in the transition to BE and to unmanaged forests (13 years). Also in the stem case, climate change does not affect the outcome of the management option with the longest $\tau_{stem}$, but in cold climates with warm summers and temperate climates without dry seasons, it makes $\tau_{stem}$ in the unmanaged forests clearly the shortest. In the stem cases, the longer $\tau_{stem}$ is mainly driven by an increase in the stem C pool rather than changes in the outflows (S17 Fig and S20 Fig). Notable exceptions are the increase of mortality in the unmanaged forests, especially in cold climates, and in BD forests in cold climates with cold summers.

In the soil, in the management-only simulations $\tau$ is similar among the different management options in each climatic zone, with the exception of the unmanaged forests, which show the shortest $\tau_{soil}$ in all the climates (Fig 5a). The longest $\tau_{soil}$ is observed in cold climates with cold summers for all the management types, while the shortest is observed in the temperate climate with dry summers. In the case of soils, differences in $\tau$ are driven primarily by outflows through heterotrophic respiration (S16 Fig). These outflows are generally the highest in unmanaged forests, with the exception of cold climates. By contrast, the size of the soil carbon pool is broadly similar across management options, with the only exception of cold climates with cold summers, where unmanaged forests exhibit the lowest soil pool, whereas BD

forests stored the largest (S15 Fig). When climate change is included (Fig 5b), as for the $\tau_{eco}$, $\tau_{soil}$ decreases in all the types of climates and for all the management options. The greatest changes are again observed in cold climates with cold summers, where the differences in $\tau_{soil}$ with the management-only simulations span between almost 20 years (in the transition to BD forests) and 40 years (in the baseline and the transition to NE forests) (Fig 5c). Other relevant decreases in $\tau_{soil}$ are projected in the cold climate with warm summers (almost 10 years for all the management options). In the case of the cold climate with cold summers, this decrease of $\tau_{soil}$ is the result both of a decrease in the soil C pool and of the aforementioned increase in heterotrophic respiration, while in cold climates with warm summer it is only due to the latter (S17 Fig and S18 Fig).

In S4 Appendix, we also present the results of simulations conducted on a subset of grid cells, where alongside the management-only and SSP3-RCP7.0 climate-forced scenarios, we included simulations driven by climate data from the SSP1-RCP2.6 scenario.

## Differences in $\tau$ for each forest type transition

We then computed the differences in $\tau$ between each management option and the baseline scenario for locations with a particular dominant stand type (either needle-leaved or broadleaved) before the beginning of the management (Fig 6). Since the transition to BE forest does not fulfil our forest definition (the 5-meter height threshold, see methods) in a great part of central and northern Europe, its extent is much smaller than the forest extent in the baseline scenario, making their averages not comparable. For this reason, we excluded the toBE scenario from this analysis.

The type of forest present before management begins - whether broadleaved-dominated or needle-leaved-dominated - produces similar results to those observed across different climatic zones in management-only simulations (Fig 6a). Specifically, transitioning to unmanaged and BD forests results in a longer $\tau_{eco}$ compared to the baseline, while transitioning to NE forests leads to a shorter $\tau_{eco}$, regardless of the initial dominant forest type. In contrast, $\tau_{stem}$ shows more pronounced differences between the two initial forest types. Transitioning to NE forest increases $\tau_{stem}$ more if the initial forest is dominated by broadleaved species (16 years longer) rather than by needle-leaved species (6 years longer). Conversely, transitioning to BD forests decreases $\tau_{stem}$ less if starting from a broadleaved-dominated forest (2 years shorter) compared to a needle-leaved-dominated forest (11 years shorter). The unmanaged forests, instead, show a longer $\tau_{stem}$ than the baseline in broadleaved-dominated forests (6 years) and a shorter $\tau_{stem}$ in needleleaved-dominated forests (4 years). The analysis of $\tau_{soil}$ reveals the greatest changes in the unmanaged forest, which exhibits a 5-year shorter $\tau_{soil}$ in broadleaved-dominated forests, but even more so in the needle-leaved-dominated forests (15 years shorter). Also a transition to BD forests leads to a shorter $\tau_{soil}$, but only starting from needle-leaved-dominated forests (-8 years). Climate change does not significantly affect the differences in $\tau_{eco}$ and $\tau_{soil}$ relative to the baseline in broadleaved-dominated forests, but affects the transition to BD forests in needle-leaved-dominated forests: $\tau_{eco}$ gets even longer than in the baseline scenario (5 years longer than in the management-only simulations) and $\tau_{soil}$ transitions from 8 years shorter to 3 years longer (Fig 6b, 6c). Climate change has instead a great impact on $\tau_{stem}$ in broadleaved-dominated stands, favouring the transition to NE forests (29 years longer than the baseline) and disadvantaging the transition to unmanaged, which has a 12-years shorter $\tau_{stem}$ than the baseline (Fig 6b, c). In needle-leaved-dominated forests, the pattern is similar, but also a transition to BD forests causes a shorter $\tau_{soil}$ (-26 years) compared to the baseline. In general, although the magnitude of changes varies slightly between the

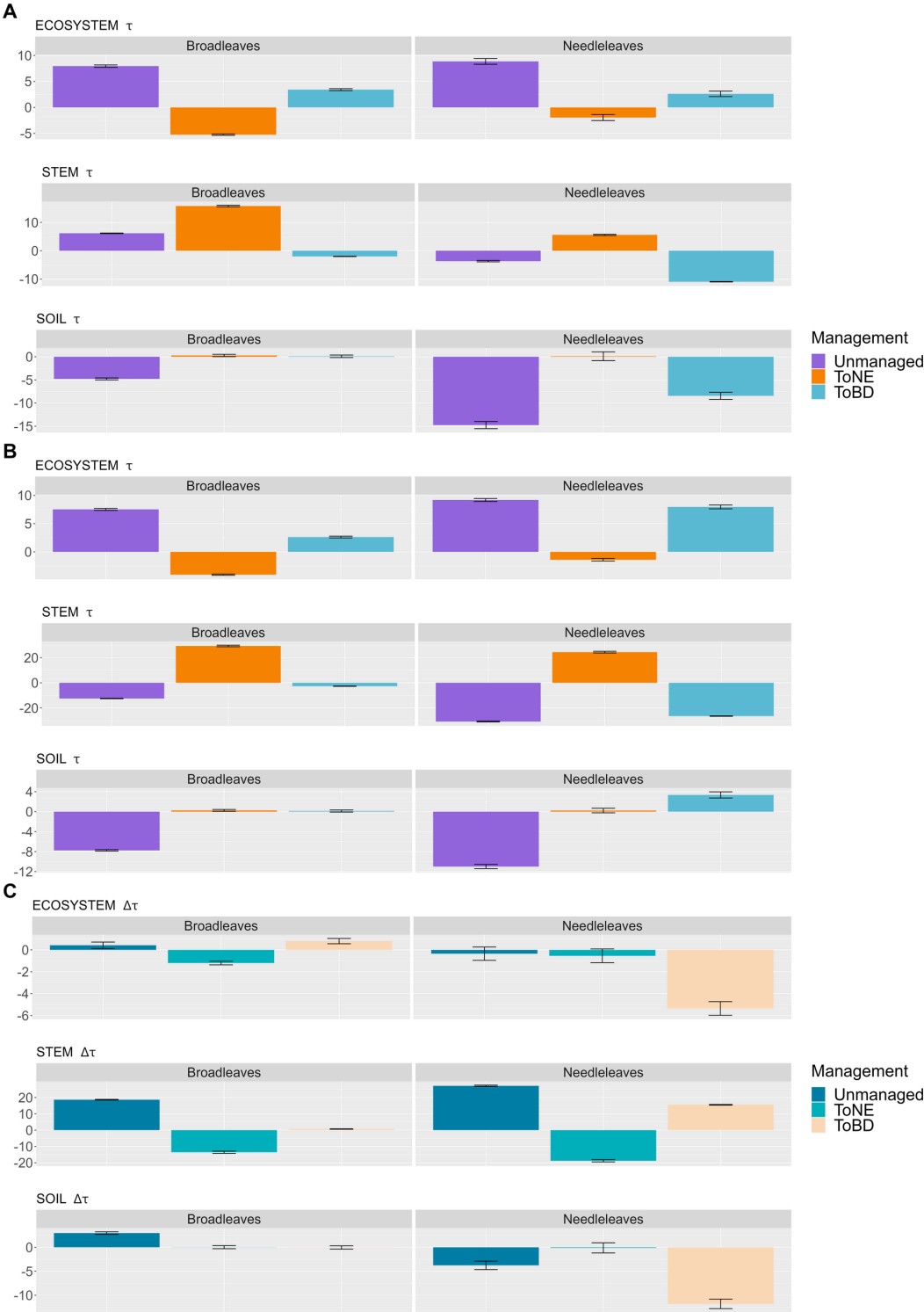

**Fig 6. Difference in the mean $\tau$ (with standard error bars) between each management option and the baseline scenario, when transitioning from broadleaved-dominated and needle-leaved-dominated grid cells, for the ecosystem, the stem pool and the soil pool.** A: in the management-only simulations; B: in the management and climate simulations; C: difference between A and B. The mean refers to the last 30 years of the 3$^{rd}$ management cycle, excluding the final clear-cut

needle-leaved and the broadleaved-dominated forests across all management options, the differences are much smaller than those observed among the climatic zones.

## Discussion

Modelling how European forests have been and are currently managed remains a challenge because a constellation of different forest owners and different forest regulations in each country creates a very mixed picture [52]. Besides this, data on management practices are rarely available, and forest management is generally inferred from remote sensing data [53] or from harvest data reported in the national forest inventories [54]. A recent study indicated that while there has been an overall increase in Net Primary Productivity (NPP) in European forests, the existing management practices may lead to a decline in carbon sink capacity by 2050 [55]. This decline is contrary to the goals necessary for Europe to meet its climate targets [56]. Given the diverse historical management patterns and the uncertainty on how management and climate change will interact in European forests over the coming decades, we chose here a modelling approach based on stylised scenarios, to explore how species composition affects $\tau$.

### The isolated effect of forest management

The results of the management-only simulations clearly show that changing species composition has an important effect on the ecosystem and stem $\tau$, while not so much on soil $\tau$. Across climates, unmanaged forests consistently exhibit the longest $\tau_{\mathrm{eco}}$. Similar to our results, Wang et al. (2018) found for forests in China, significantly longer $\tau$ in natural *vs* planted forests [57]. Although our "unmanaged" forests are not pristine, but rather transition to unmanaged forests, we can assume that after more than 200 years (at the time of the calculation of $\tau$), the structure of the unmanaged forest is comparable to a natural forest. In unmanaged forests, fire fluxes are included in the calculation of $\tau_{\mathrm{eco}}$, but their contribution is generally small, with the exception of cold climates ( S21 Fig). Nevertheless, $\tau_{\mathrm{eco}}$ remains the longest also in these regions. Other studies suggested that an increasing share of unmanaged forests is essential to maintain European forests as carbon sinks, with co-benefits in terms of biodiversity and trade-offs with timber production [28,58]. In our simulations, unmanaged forests are best compared with the baseline scenario, since the only structural difference (besides the presence of fire) lies in the absence of the 80-year rotation cycle (resulting in uneven-aged stands). A consistent outcome is that unmanaged forests hold a larger total carbon pool than the baseline, which aligns with empirical evidence showing that uneven-aged and unmanaged stands tend to accumulate greater above- and below-ground biomass over extended periods [22,59–61]. At the same time, unmanaged forests exhibit lower autotrophic respiration compared to the baseline. A study from Collalti et al. (2018) also simulated a lower autotrophic respiration in unthinned *vs* thinned forests [62]. We are not aware of other studies comparing autotrophic respiration in uneven *vs* even-aged forests, or between unmanaged *vs* managed forests. Heterotrophic respiration is instead generally higher in unmanaged forests than in the baseline. As also suggested by Luyssaert et al. (2007) and Harmon et al. (2011), this can be attributed to the accumulation of litter and coarse woody debris [63,64], which in our simulations are partly removed in managed systems during the harvest. This is also reflected by the higher litter pools observed in the unmanaged forests. In cold climates, this pattern is reversed, and heterotrophic respiration, as well as the litter pools, are not consistently higher in unmanaged stands in these regions. A possible explanation is that in these unmanaged forests, fire fluxes are significant, directly combusting organic matter and reducing the substrate for microbial decomposition.

Another important result is the longer $\tau_{\text{eco}}$ of BD forests compared to the baseline and NE forests. The longer $\tau_{\text{eco}}$ is due to higher total carbon pools and lower outflows (driven by the lower autotrophic respiration). This is in line with the recent finding of Luo et al. (2024), who reported a higher carbon use efficiency in broadleaved deciduous forests compared to needle-leaved evergreen and broadleaved evergreen forests, irrespective of the type of climate [65]. This result gives another reason to lean towards a greater share of broadleaved species as a win-win solution in terms of climate change mitigation and adaptation. An increased share of broadleaved species - especially in needle-leaved-dominated forests - has been found to maximise ecosystem services in European forests [28], reduce the risk of fire [66] and increase the surface albedo [67] in boreal forests, and lead to positive effects in terms of biodiversity in coniferous monocultures [68].

$\tau_{\text{stem}}$ is the longest in unmanaged forests only in arid and temperate climates with dry summers, while in other climatic zones, NE forests exhibit the longest $\tau_{\text{stem}}$. The general pattern is that unmanaged forests accumulate greater C stem pool, but at the same time, exhibit significantly higher tree mortality compared to managed forests, consistent with findings by Montoya et al. (2023) [69] and Kern et al. (2021) [70]. Despite in managed forests thinning adds up to the outflows, it does not compensate for the higher mortality in unmanaged forests. $\tau_{\text{stem}}$ outcome is then not as straightforward as for $\tau_{\text{eco}}$. NE forests always display, among the managed forests, the longest $\tau_{\text{stem}}$ due to their lower background mortality and reduced thinning harvest intensity. The low thinning intensity is a feature of the simulation setup, whereas the lower mortality is consistent with the slower growth and developmental stage of NE forests. At 50–80 years post-clearcut, NE forests often remain in an early competition phase, while BD forests—with faster growth—enter competitive self-thinning earlier, resulting in higher mortality [71].

In our simulations, all managed plantation scenarios exhibit very similar $\tau_{\text{soil}}$, whereas unmanaged forests consistently show shorter $\tau_{\text{soil}}$ and higher heterotrophic respiration. This pattern suggests that, within the mid-rotation window we analyse, factors other than a change in species composition—such as forest structure and, in the cold climates, the role of fire—dominate $\tau_{\text{soil}}$ dynamics. Indeed, Wang et al. (2021) found no significant differences in microbial respiration between broadleaved and needleleaved species, or between deciduous and evergreen types, when converting forests to plantations, supporting the interpretation that species identity exerts only limited control over $\tau_{\text{soil}}$ at this timescale [57]. Short-term increases in soil respiration following clear-cut have been well documented, but these pulses generally attenuate over subsequent decades [72–74]. By contrast, unmanaged stands can sustain greater inputs of labile substrate—through coarse woody debris [75], and exhibit more heterogeneous spatial structure, with thicker and more disturbed soil horizons [76] that can stimulate microbial activity.

**The peculiar case of the cold climate with cold summers** In our experiments, the cold regions with cold summers exhibit distinct trends compared to the rest of Europe, characterised by greater differences between the management options and contrasting trends in $\tau_{\text{stem}}$ for unmanaged forests relative to the baseline scenario. These differences can be attributed to various factors. Of all the regions, this is the one where forests deviate the most from the equilibrium (see S2 Appendix), meaning that the calculation of $\tau$ likely is the least accurate (because the turnover *rate* could change during the years, consequently changing $\tau$). In this climate, in fact, the correlation between $\tau_{\text{eco}}$ calculated with the outflows compared with $\tau_{\text{eco}}$ calculated based on Gross Primary Productivity (GPP) is the weakest, especially when considering the unmanaged forests and the transition to BD forests (although the values of the Spearmann correlation coefficients are still acceptable (0.94 and 0.96, respectively S5 Fig)). For the unmanaged forests in this region, the influence of forest structure and fire (the only

differences from the baseline) becomes evident. $\tau_{\text{stem}}$ is shorter than the baseline, unexpectedly so given the absence of carbon loss from harvesting activities. Here, the presence of fire in the unmanaged forests plays a role in calculating $\tau$, particularly for $\tau_{\text{stem}}$ (and less so for $\tau_{\text{eco}}$, where respiration is dominant), while presence or absence of fire does not alter $\tau$ notably in the other climates. Losses due to fires represent, in fact, a substantial outflow in cold climates with cold summers when it comes to $\tau_{\text{stem}}$ (40% of the total outflows) (S21 Fig). This partially explains the apparent counterintuitive shorter $\tau_{\text{stem}}$ in unmanaged forests. Besides this, in the northern countries, the age distribution in unmanaged forests diverges from the rest of Europe, showing a greater density of young trees (<50 years) (S22 Fig). Together with the fire, this could also be one of the reasons for the lower heterotrophic respiration in the unmanaged forests. Tang et al.(2008) found in fact a lower soil respiration in younger forest stands compared to mature stands [77], and Pregitzer and Euskirchen (2004) observed, in boreal forests, a higher heterotrophic respiration in young trees (30-70 years) than in trees belonging to the following class (70 - 120 years) [78]. Although definitions of young, mature, and old forests vary across studies, making direct comparisons challenging, it is clear that differences in forest structure play a critical role in shaping forest fluxes [79]. The importance of past forest management in shaping the forests and determining their carbon sink is highlighted by Forkel et al. (2019) [20]. In the northern countries, characterised by heavy management before 2010, forest density is higher than in the rest of Europe (justifying also the greater impact of fire), and the simulations start from a forest that comprises a much higher proportion of younger trees, leading to different mortality patterns after 210-240 years — when $\tau$ calculations are made.

## The combined effects of forest management and climate change

Adding the effect of climate change and increasing carbon dioxide ($CO_2$) concentration produces different impacts on $\tau$ of different pools. $\tau_{\text{eco}}$ and $\tau_{\text{soil}}$ shorten compared to the management-only simulations. This reduction is consistent across all forest management options, indicating that climate change impacts these pools similarly regardless of management. Yan et al. (2017), likewise observed declining mean $\tau$ for both soil and ecosystem carbon between 1901 and 2011, and showed a negative exponential relationship between mean temperature and $\tau$ [80]. The reason behind the decreasing trend in $\tau_{\text{eco}}$ in our simulations is an increase in the outflows. In particular, we observe an increase in autotrophic respiration (the main contributor to the calculation of $\tau_{\text{eco}}$), in line with its well-known temperature sensitivity [81]. Although the total C pools also tend to increase, this increase is not sufficient to compensate for the higher autotrophic respiration. Similarly, Collalti et al. (2018), in another modelling experiment, found that autotrophic respiration rates increase more strongly than photosynthetic rates under warming conditions [62]. The magnitude of respiration increase varies in our simulations: it is more pronounced for baseline and NE forests in cold climates, and for evergreen forests in other climate zones. This dynamic further strengthens the relative advantage of unmanaged and BD forests under future climate conditions. Among climate zones, the cold regions with cold summers experience the steepest decreases in $\tau_{\text{eco}}$ (exceeding 20 years across all management options). Yan et al. (2017) also reported that boreal regions exhibit the strongest decline in turnover times, driven by stronger-than-average warming compared to the global mean of +1 °C during their study period [80]. Yet, the story changes when examining the stem carbon pool. Here, most management scenarios and climates see a lengthening in $\tau_{\text{stem}}$, as stem biomass accumulates and outflows remain stable. However, unmanaged forests in cold climates and temperate climates without dry seasons are a major exception. In these systems, mortality rates increase markedly, shortening $\tau_{\text{stem}}$

and reflecting what was modelled by Yu et al. (2019): greater growth is paired with greater mortality, leading to shorter above-ground vegetation $\tau$ in boreal and temperate forests [14]. This dynamic matches the "grow fast – die young" hypothesis described by Büntgen et al. (2019) [7]. If trees are growing faster due to resource enrichment (e.g. carbon dioxide ($CO_2$) or nitrogen deposition), they may also die younger, ultimately preventing forests from accumulating more long-term carbon [7]. On the other hand, our simulations in managed forests always predict a longer $\tau_{stem}$ compared to the management-only simulations. Thinning practices, by reducing stand density and competition, can make resources such as water and light become more available to the remaining trees, which in turn helps them withstand climate stresses like drought or heat waves [82,83]. We hypothesize that a comparable effect could explain patterns in our simulations, though we are unable to test this explicitly. An important shortcoming in this respect, is mortality due to disturbance, which in our study does not change as a consequence of climate change and/or management, and probably underestimates the outflows in the future projections and could consequently overestimate $\tau_{stem}$ calculation (see Uncertainties and future work). The fact that the greatest differences between the management-only and the management and climate change simulations are observed in areas of cold climates with cold summers and, to a lesser extent, of cold climates with warm summers, has two main explanations. First, the increase in temperature is predicted to be higher here, compared to the rest of Europe. This causes a change in the climatic zones identified in Beck et al. (2023) [51]. By the end of the century in fact, projections point to a shift from a cold climate with cold summers to a cold climate with warm summers in northern Sweden and Finland, and a shift from a cold climate with warm summers to a temperate climate without dry season in the southern part of these countries (S11 Fig). Second, as for the management-only simulations, the forest is not in full equilibrium in this area (S6 Fig), especially in the unmanaged and BD forests. Here, $\tau_{eco}$ calculated with the Gross Primary Productivity (GPP) equation is shorter than $\tau_{eco}$ calculated with the outflows, indicating an ongoing C accumulation.

## Uncertainties and future work

Our results need to be interpreted in light of important limitations, some of which could be addressed in future studies. First of all, given the intrinsic uncertainties in the mechanisms that influence the modelling of $\tau$, more robust conclusions could be drawn in a study that would utilise Dynamic Global Vegetation Model (DGVM)s with different representations of mortality and allocation (such as in Pugh et al. 2020 [19]) and the capability of representing different harvest options. This would allow not only to assess the uncertainties but also to better understand the mechanisms that drive the changes between the different forest management options.

Second, the coarse resolution of $\tau$ calculation in our simulations (0.5°x 0.5°) could be subjected to biases. Climate data at a higher resolution (0.05°x 0.05°) are now available for Europe [84]. Otryakhin et al. (2025) reported that in LPJ-GUESS, biases in C pools and fluxes due to the coarse resolution of the climatic data do exist, but they are limited to mountainous areas, coastlines and inland water bodies [84]. Given the stylised nature of our simulation, the continental-scale of the study, and the absence of high-resolution datasets of forest management practices at European level, the coarse resolution of our $\tau$ calculation seems reasonable. However, a higher resolution would be recommended in smaller-scale studies, in particular if focused on the Alpine region (where our simulations often report the greatest differences between the management options).

The third important shortcoming is the implementation of disturbance. This study does not consider potential changes in the frequency of disturbances based on species composition. Needle-leaved species are generally more susceptible to disturbances [28,85,86], particularly stand-replacing disturbances [87]. Conversely, broadleaved species may be more susceptible to localised events such as ice and frost [88], which in Europe are expected to decrease with climate change [85]. This is in contrast to disturbances like fires, droughts, windthrows, and insect outbreaks, which are predicted to increase in European forests [85]. These differences in disturbance susceptibility are not represented in our simulations and could increase even more the gap in $\tau_{eco}$ between unmanaged and broadleaved forests *vs* needle-leaved forests. Our current results may in fact overestimate $\tau_{eco}$ in the NE forests, leading to a conservative bias in our conclusions. Incorporating species-specific disturbance susceptibility in future studies, would likely improve the ecological realism of $\tau_{eco}$ projections.

Finally, another source of uncertainty arises from the choice of rotation length: in this study, we selected 80 years, which is somewhat arbitrary but falls within the range typically applied in European forests. This decision could influence the differences observed between broadleaved and needle-leaved species. For instance, Pilli et al. (2022) [55] found that conifers maintain stable net ecosystem production (GPP minus heterotrophic respiration) until 2070 under current forest management practices, whereas broadleaved forests show a decline. This decline is attributed to the ageing of broadleaved forests, which are managed differently compared to needle-leaved forests in their study, reflecting more realistic and country-specific management options. As a result, the net biomass accumulation rate for broadleaved species decreases over time. Therefore, besides species composition, other management practices (thinning frequency and intensity, clear-cut rotation length) should be explored. We also tested a rotation time of 60 and 100 years on a limited amount of grid cells (S10 Fig), which had some small effects on $\tau_{stem}$ but did not alter the main conclusions; however, further combinations should be explored to identify the optimal strategy for maximising $\tau$. While this may not be feasible at a European scale, it could be implemented in smaller regions with similar forest management practices. This study used, in fact, homogeneous and stylised scenarios applied across Europe, to investigate the consequences on $\tau$ of actively shifting species composition. The reality of European forests—shaped by diverse management histories and varying intensities of climate change—requires management strategies that are sensitive to local contexts if they are to be effective for both mitigation and adaptation and to contribute to achieving the European climate targets.

## Conclusion

This modelling study emphasises species composition as a key variable and establishes a basis for integrating a range of forest management scenarios into the calculation of $\tau$, with the aim of improving projections of future carbon stocks in European forests. Notable uncertainties persist, particularly related to disturbance regimes, rotation lengths, and the post-harvest fate of wood products. Further research should focus on reducing these uncertainties and evaluating region-specific management strategies to optimise carbon outcomes under varying climate change contexts. From a policy perspective, given the stylised scenarios that were applied to the whole continent, more tailored regional studies that include other options (different rotation lengths, mixed forests, other thinning strategies), would make our findings more relevant for concrete goals in the e.g. Land Use, Land-Use Change, and Forestry (LULUCF) sector.

Despite the aforementioned limitations, our study gives some general indications. First, the model results show that switching to unmanaged forests or broadleaved forests can lengthen

$\tau_{\text{eco}}$, supporting climate mitigation. Second, $\tau_{\text{stem}}$ shows different patterns compared to $\tau_{\text{eco}}$ and $\tau_{\text{soil}}$, highlighting the importance of including the entire forest ecosystem and not only above-ground woody biomass when calculating $\tau$, if the aim is to assess measures to mitigate climate change.

## Acknowledgments

We thank Paul Miller, Stefan Olin, Karl Piltz and Susanne Suvanto for developing the version of LPJ-GUESS on which this study is based.

## Supporting information

**S1 Appendix. Changes in the nitrogen parameters.**
(PDF)

**S2 Appendix. The equilibrium assumption.**
(PDF)

**S3 Appendix. $\tau$ of the soil pools.**
(PDF)

**S4 Appendix. Comparison of $\tau$ among different climate scenarios and with different rotation lengths.**
(PDF)

**S1 Fig. Relative difference between the results of $\tau_{\text{eco}}$ calculated based on the outflows and on the GPP, for the management-only simulations.** From light green to dark blue, the absolute relative difference increases. The colour scale only shows negative values because no cases were observed where the results of the outflow calculation were greater than the results of the GPP calculation. Areas where the forest does not reach an average tree height of 5 meters are excluded.
(TIFF)

**S2 Fig. Relative difference between the results of $\tau_{\text{eco}}$ calculated based on the outflows and on the GPP, for the management and climate change (SSP 3–7.0) simulations.** Colour coding as in S1 Fig. Areas where the forest does not reach an average tree height of 5 meters are excluded.
(TIFF)

**S3 Fig. Paired boxplots of $\tau_{\text{eco}}$ results calculated with the different equations, for the management-only simulations.** The results refer to each management option within each climatic zone. In pink, results obtained with the outflow equation and in purple, results obtained with the GPP equation.
(TIFF)

**S4 Fig. Paired boxplots of $\tau_{\text{eco}}$ results calculated with the different equations, for the management and climate change simulations.** Colour coding as in S3 Fig.
(TIFF)

**S5 Fig. Correlation between the results of $\tau_{\text{eco}}$ calculated with the outflow equation (x axis) and with the GPP equation (y axis), for the management-only simulations.** The results refer to each management option within each climatic zone. Spearman correlation coefficients ($\rho$) and p values ($p$) are indicated in each plot.
(TIFF)

**S6 Fig. Correlation between the results of $\tau_{eco}$ calculated with the outflow equation (x axis) and with the GPP equation (y axis), for the management and climate change simulations.** The results refer to each management option within each climatic zone. Spearman correlation coefficients ($\rho$) and p values ($p$) are indicated in each plot.
(TIFF)

**S7 Fig. Differences in $\tau_{surfhumus}$ between each management option and the baseline scenario, in the management-only simulations - 30-year average before the last clear-cut.** Colour coding as in Fig 2. Areas where the forest does not reach an average tree height of 5 meters are excluded.
(TIFF)

**S8 Fig. Differences in $\tau_{surfhumus}$ between each management option and the baseline scenario, in the management and climate change simulations - 30-year average before the last clear-cut.** Colour coding as in Fig 2. Areas where the forest does not reach an average tree height of 5 meters are excluded.
(TIFF)

**S9 Fig. Mean $\tau$ for each management type, in different climatic zones, for the ecosystem (A), the stem (B) and the soil (C) pool, calculated for a small sample of grid cells.** The labels on the bars indicate the climate data used: No CC = management-only simulations, rcp 2.6 = SSP1-RCP2.6 climate change scenario, rcp 7.0 = SSP3-RCP7.0 climate change scenario. The mean refers to the last 30 years of the 3rd management cycle, excluding the final clear-cut in the management-only simulations, and to the 2060–2089 average in the two SSPs-RCPs simulations.
(TIFF)

**S10 Fig. Mean difference in $\tau$ between the transition to NE, BD and BE forests and the baseline option, in different climatic zones, for the ecosystem (A), the stem (B) and the soil (C) pool, calculated for a small sample of grid cells.** The labels on the bars indicate the rotation length (60 - 80 and 100 years). The mean refers to the last 30 years of the 3rd management cycle, excluding the final clear-cut in the management-only simulations.
(TIFF)

**S11 Fig. European climatic zones, grouped from [51] for the period 1991 - 2020 (panel A on the top) and the period 2071 - 2099 according to the SSP3-RCP7.0 (panel B at the bottom).** Abbreviations as per the Köppen-Geiger classification: BWh = Arid, desert, hot; BSh = Arid, steppe, hot, BSk = Arid, steppe, cold; Csa = Temperate, dry summer, hot summer; Csb = Temperate, dry summer, warm summer; Cfa = Temperate, no dry season, hot summer; Cfb = Temperate, no dry season, warm summer; Dsb = Cold, dry summer, warm summer; Dfa = Cold, no dry season, hot summer; Dfb = Cold, no dry season, warm summer; Dfc = Cold, no dry season, cold summer; ET = Polar, tundra.
(TIFF)

**S12 Fig. Dominant stand type.** Each grid cell shows the stand type with the highest forest cover percentage at the beginning of the simulations (2010): NE = needle-leaved evergreen, ND = needle-leaved deciduous, BE = broadleaved evergreen, BD = broadleaved deciduous).
(TIFF)

**S13 Fig. Relative difference in $\tau$ between each management option and the baseline scenario, for the management-only simulations.** Different shades of yellow indicate that $\tau$ is longer in the baseline than in the other management options, while different shades of blue

indicate that it is shorter. Areas where the forest does not reach an average tree height of 5 meters are excluded.
(TIFF)

**S14 Fig. Relative difference in $\tau$ between each management option and the baseline scenario, for the management and climate change simulations.** Different shades of yellow indicate that $\tau$ is longer in the baseline than in the other management options, while different shades of blue indicate that it is shorter. Areas where the forest does not reach an average tree height of 5 meters are excluded.
(TIFF)

**S15 Fig. Trends in the C pools for the period of $\tau$ calculations in the management-only simulations (end of the 3rd cycle).** Different colours indicate the different management options, while different line types indicate the different pools.
(TIFF)

**S16 Fig. Trends in the ecosystem C fluxes for the period of $\tau$ calculations in the management-only simulations (end of the 3rd cycle).** Different colours indicate the different management options, while different line types indicate the different fluxes.
(TIFF)

**S17 Fig. Trends in the ecosystem C pools in the management and climate change simulations.** Different colours indicate the different management options, while different line types indicate the different pools. The vertical red line highlights the beginning of the management, and the vertical black line indicates the end of the 1st rotation period.
(TIFF)

**S18 Fig. Trends in the ecosystem C fluxes in the management and climate change simulations.** Different colours indicate the different management options, while different line types indicate the different fluxes. The vertical red line highlights the beginning of the management, and the vertical black line indicates the end of the 1st rotation period.
(TIFF)

**S19 Fig. Trends in the stem C fluxes for the period of $\tau$ calculations in the management-only simulations (end of the 3rd cycle).** Different colours indicate the different management options, while different line types indicate the different fluxes.
(TIFF)

**S20 Fig. Trends in the stem C fluxes in the management and climate change simulations.** Different colours indicate the different management options, while different line types indicate the different fluxes. The vertical red line highlights the beginning of the management, and the vertical black line indicates the end of the 1st rotation period.
(TIFF)

**S21 Fig. Mean outflow contribution in the calculation of A) $\tau_{eco}$ and B) $\tau_{stem}$ in the management-only simulations, for each management type, in different climatic zones.** The mean refers to the last 30 years of the 3rd management cycle, excluding the final clear-cut.
(TIFF)

**S22 Fig. Proportion of each forest age class in the baseline and the unmanaged forests (average 2220–2249).** A-B-C: Young forests (< 50 years), mature forests (between 51 and 140 years), and old forests (>140 years) in the unmanaged forests. D-E-F: Young forests, mature forests, and old forests in the baseline).
(TIFF)

**S1 Table. Absolute values of $\tau$ for the ecosystem, stem and soil pools in each climatic zone, for the management-only and the management and climate change simulations, further separated by management scenarios.**
(PDF)

## Author contributions

**Conceptualization:** Anna Ferretto, Peter Anthoni, Thomas A. M. Pugh, Almut Arneth.

**Data curation:** Peter Anthoni, David Wårlind.

**Formal analysis:** Anna Ferretto, Peter Anthoni.

**Funding acquisition:** Almut Arneth.

**Investigation:** Anna Ferretto.

**Methodology:** Anna Ferretto, Peter Anthoni, Thomas A. M. Pugh, Almut Arneth.

**Software:** Mats Lindeskog.

**Supervision:** Peter Anthoni, Thomas A. M. Pugh, Almut Arneth.

**Visualization:** Anna Ferretto.

**Writing – original draft:** Anna Ferretto.

**Writing – review & editing:** Peter Anthoni, Thomas A. M. Pugh, Konstantin Gregor, Martin Thurner, Carolina Natel, David Wårlind, Mats Lindeskog, Almut Arneth.

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
