## [Decision Letter · Decision Letter 0]

29 Jul 2025

PONE-D-25-28344The impact of changing species composition in Europe - longest carbon turnover time in unmanaged and broadleaved deciduous forestsPLOS ONE

Dear Dr. Ferretto,

Thank you for submitting your manuscript to PLOS ONE. After careful consideration, we feel that it has merit but does not fully meet PLOS ONE’s publication criteria as it currently stands. Therefore, we invite you to submit a revised version of the manuscript that addresses the points raised during the review process.

**Please make revisions based on the suggestions of the referees.Especially clarify the methods used,make discussions accprding to the results**

We look forward to receiving your revised manuscript.

Kind regards,

RunGuo Zang

Academic Editor

PLOS ONE

Journal Requirements:

[This study was possible thanks to funds awarded to AF from the Alexander von Humboldt Stiftung.]. 

[This study was made possible thanks to the funds granted to AF by the Alexander von Humboldt Foundation. TP and ML were funded under the ForestValue programme, the European Commission, Vinnova, the Swedish Energy Agency and Formas for the project FORECO. This study is a contribution to the Swedish government’s strategic research areas BECC and MERGE and the Nature-based Future Solutions profile area at Lund University. KG acknowledges funding from the VELUX Stiftung through the 3FOR-project (project 1897, www.velux-stiftung.ch, www.3for-project.org). We thank Paul Miller, Stefan Olin, Karl Piltz and Susanne Suvanto for developing the version of LPJ-GUESS on which this study is based.]

[This study was possible thanks to funds awarded to AF from the Alexander von Humboldt Stiftung.]

6. Please amend either the title on the online submission form (via Edit Submission) or the title in the manuscript so that they are identical.

7. We note that Figures 2, 3, 4, S1, S2, S3, S8, S9, S10, S11, S12, S14, and S15 in your submission contain map images which may be copyrighted. All PLOS content is published under the Creative Commons Attribution License (CC BY 4.0), which means that the manuscript, images, and Supporting Information files will be freely available online, and any third party is permitted to access, download, copy, distribute, and use these materials in any way, even commercially, with proper attribution. For these reasons, we cannot publish previously copyrighted maps or satellite images created using proprietary data, such as Google software (Google Maps, Street View, and Earth). For more information, see our copyright guidelines: http://journals.plos.org/plosone/s/licenses-and-copyright.

We require you to either (1) present written permission from the copyright holder to publish these figures specifically under the CC BY 4.0 license, or (2) remove the figures from your submission

1. You may seek permission from the original copyright holder of Figures 2, 3, 4, S1, S2, S3, S8, S9, S10, S11, S12, S14, and S15 to publish the content specifically under the CC BY 4.0 license. 

8. We notice that your supplementary figures are uploaded with the file type 'Figure'. Please amend the file type to 'Supporting Information'. Please ensure that each Supporting Information file has a legend listed in the manuscript after the references list.

9. We notice that your supplementary tables are included in the manuscript file. Please remove them and upload them with the file type 'Supporting Information'. Please ensure that each Supporting Information file has a legend listed in the manuscript after the references list.

Additional Editor Comments:

Please make revisions according to the concerns of the referees

Reviewers' comments:

Reviewer's Responses to Questions

**Comments to the Author**

1. Is the manuscript technically sound, and do the data support the conclusions?

Reviewer #1: Yes

Reviewer #2: Yes

2. Has the statistical analysis been performed appropriately and rigorously? 

Reviewer #1: Yes

Reviewer #2: Yes

3. Have the authors made all data underlying the findings in their manuscript fully available?

Reviewer #1: Yes

Reviewer #2: Yes

4. Is the manuscript presented in an intelligible fashion and written in standard English?

Reviewer #1: Yes

Reviewer #2: Yes

5. Review Comments to the Author

Reviewer #1: PONE-D-25-28344

The manuscript presents a well-structured and methodologically rigorous study on carbon turnover time (τ) in European forests under different management and climate scenarios. The research addresses a critical gap in understanding how forest management strategies influence carbon sequestration potential, with implications for climate mitigation policies. The use of the LPJ-GUESS model and stylized management scenarios is appropriate, and the results are clearly presented. However, some sections could benefit from clarification, and limitations should be more explicitly discussed.

- Clearly states the research objective, methodology, and key findings.

- Highlights the significance of unmanaged and broadleaved deciduous forests for carbon turnover.

- Briefly mention the spatial variability of results (e.g., cold vs. temperate climates) to better contextualize the findings.

In introduction, clarify why LPJ-GUESS was chosen over other DGVMs (e.g., its management module). Expand on the rationale for selecting SSP3-RCP7.0 (e.g., its relevance to current European policy scenarios).

In M&M, provide more detail on the "stylized management options" (e.g., how they reflect real-world practices).

Discuss potential biases from excluding fire in managed forests (mentioned briefly but warrants deeper analysis).

In result section, include a summary table of mean values across all scenarios for quick comparison.

Discuss the ecological mechanisms behind shorter τ in unmanaged forests’ soil (e.g., younger stand age, decomposition rates).

In discussion, address the counterintuitive finding of longer τ in unmanaged forests despite younger stands (link to disturbance regimes or model assumptions).

Expand on the policy implications (e.g., how findings align with EU carbon neutrality goals).

Discuss how the exclusion of species-specific disturbance susceptibility (e.g., conifers vs. broadleaves) might skew results.

Address potential overgeneralization due to the 0.5° resolution (e.g., local management heterogeneity).

In conclusion, recommend specific future research directions (e.g., integrating regional management practices) at the end.

Improve the quality of figures,

label panels in Figures 2–3 more clearly (e.g., "A: Ecosystem τ").

Include a map of Europe in the main text to orient readers to climatic zones.

Define acronyms (e.g., DOC, SOM) at first use.

Ensure consistency in terminology (e.g., "broadleaved" vs. "broadleaf").

Reviewer #2: This study provides a valuable contribution to understanding carbon turnover dynamics in European forests under different management scenarios and climate change. The use of LPJ-GUESS to model carbon pools and fluxes is robust. However, several issues must be addressed before the paper can be accepted.

1. The authors chose SSP3-RCP7.0 scenario in the climate change simulation. Why not also include a moderate (SSP2-4.5) scenario and compare the results? Please discuss how a lower-forcing scenario could alter the results.

2. Evidence shows managed forests still burn. Explain why fire is turned off in managed forests.

3. The 80-year rotation cycle is described as arbitrary but reasonable. However, it varies significantly across different vegetation types and geographical regions. A supplementary run with 60- and 100-year rotations would strengthen robustness.

4. The stylised planting rules (immediate species switch after death) may overestimate the rate of species conversion. Discuss how this influences τ differences.

5. Nitrogen limitation is explicitly modelled, but no result is shown.

6. The divergent responses of τeco (decrease) and τstem (increase) under climate change need more mechanistic explanation.

7. It would be beneficial to further elaborate on how these results could influence forest management practices in Europe.

8. Please enhance the resolution of the images to improve their clarity.

6. PLOS authors have the option to publish the peer review history of their article (what does this mean?). If published, this will include your full peer review and any attached files.

Reviewer #1: **Yes: **Khawaja Shafique Ahmad

Reviewer #2: No

---

## [Author Response · Author response to Decision Letter 1]

22 Sep 2025

We attached the Response to the reviewers' comments.

---

## [Editor Report · Decision Letter 1]

23 Sep 2025

The impact of changing forest composition in Europe - longest carbon turnover time in unmanaged and broadleaved deciduous forests

PONE-D-25-28344R1

Dear Dr. Ferretto,

We’re pleased to inform you that your manuscript has been judged scientifically suitable for publication and will be formally accepted for publication once it meets all outstanding technical requirements.

Kind regards,

RunGuo Zang

Academic Editor

PLOS ONE
---

## [Editor Report · Acceptance letter]

PONE-D-25-28344R1

PLOS ONE

Dear Dr. Ferretto,

I'm pleased to inform you that your manuscript has been deemed suitable for publication in PLOS ONE. Congratulations! Your manuscript is now being handed over to our production team.

Kind regards,

on behalf of

Professor RunGuo Zang

Academic Editor

PLOS ONE